# Maternity waiting homes utilization and associated factors among women who gave birth in the last one year in rural settings of Basona Worena District, Ethiopia: A cross sectional study

Endale Menkir Degife[1], Eyosiyas Yeshialem[2], Abdurrahman Mahammed Ahmed[2], Taye Anbessie Teklemariam[3], Abebe Nigussie Ayel [4]*

1 Department of public health, Basona worena District Health Office, Ethiopia, 2 Department of Public Health, Debre Berhan University, Asrate Woldeyes Health Science Campus, Ethiopia, 3 Department of Public Health, Community Health insurance, Ethiopia, 4 Department of Pediatrics Nursing, Debre Berhan Health Science College, Ethiopia

* abebe2014nigussie@gmail.com

## Abstract

### Background

Maternal waiting home is a residence near to health centers or hospitals that can be used as a temporary house for pregnant women for several days, while waiting for delivery reached, and a few days after labor. Most of the scholars focused on assessing the intention and knowledge of mothers to utilize maternal waiting homes for their recent delivery even though ignorance of utilization. In Ethiopia, the utilization of maternal waiting homes and its associated factors among women who gave birth in rural setting were not clearly described.

### Objectives

The overall objectives of this study were to assess maternity waiting home utilization and associated factors among women who gave birth in the last one year in the rural settings of Basona Worena District, Ethiopia, in 2024.

### Methods

A community-based cross-sectional study was conducted in Basona worena district. Multi-stage sampling techniques were used to select 460 study participants. Structured and pre-tested interviewer-administered questionnaires were used to collect data. Data were entered to Epi-data version 4.6 and exported to SPSS version 25 software for cleaning and statistical analysis. Bivariable and multi-variable logistic regression analysis was conducted to identify the association between dependent and independent variables and strength of association was measured based AOR with 95% confidence interval. Statistical significance was declared at p-value less than 0.05.

**Data availability statement:** All data are in the manuscript and/or supporting information files.

**Funding:** The author(s) received no specific funding for this work.

**Competing interests:** The authors declare that they have no competing interests exist.

**Abbreviations: ANC**, Ante Natal Care; **AOR**, Adjusted Odd Ratio; **BEMONC**, Basic Emergency Obstetrics and New born Care; **CI**, Confidence Interval; **EDHS**, Ethiopian Demographic Health Survey; **HF**, Health Facility; **HIV**, Human Immune Virus; **MM**, Maternal Mortality; **MMR**, Maternal Mortality Rate; **MWH**, Maternal Waiting Home; **PNC**, Post Natal Care; **SD**, Standard Deviation; **SDG**, Sustainable Development Goal; **SPSS**, Statistical Package for Social Sciences; **SSA**, Sub-Saharan Africa; **TBA**, Traditional Birth Attendant; **WHO**, World Health Organization.

## Result

The overall magnitude of maternity waiting home utilization was 56.7% (95% CI: 52.4, 61.3). In this study, family size (AOR = 2.76, 95%, CI: 1.27,5.99), government-employed women(AOR = 0.12,95%,CI:0.03,0.44),maternal age (26–30years) (AOR = 0.22,95% CI:0.08,0.65), primary level maternal education (AOR = 3.20,95%,CI:1.40,7.32), birth preparedness plan (AOR = 10.23,95%,CI:9.8,29.3), and MWH utilization plan (AOR = 6.82,95%,CI: 2.7,17.3) were significantly associated with maternity waiting home utilization.

## Conclusion

The overall maternity waiting home utilization was 56.7%, which is relatively low compared to previous studies. Therefore, more attention is needed to improve maternal education, strengthen the birth preparedness plan, and MWH utilization plan, as well as focus high-parity women on their birth complications readiness, which accelerates maternity waiting home utilization.

## Introduction

Maternal waiting home (MWH) is a residence near to health centers or hospitals that can be used as a temporary house for pregnant women for a number of days, while waiting delivery reached, and a few days after labor [1].MWH increases access to ANC visits, postnatal care and health information about family planning and child vaccination by a health professional [2]. It is a highly profitable and inexpensive approach to reduce maternal morbidity and mortality as well as it is a low-cost solution to access skilled birth attendants in remote areas [3].The maternity waiting home enables access to skilled care during intrapartum and postpartum periods, predominantly for women living in rural and remote areas where distance and poor transportation harshly limit access to birth services [4].

Utilization levels of MWHs globally have generally been described to be low with their conditions often regarded as insufficient [5]. Maternal mortality still a global problems and nearly 830 women die due to pregnancy and child birth every day in the world, of whom 99% are in Sub-Saharan countries [6]. Currently,MMR in Ethiopia is still high, 305 per 100,000 live births [7].United Nations sustainable development goal three target one plan, the global maternal mortality ratio will be less than 70 per 100,000 live births by 2030 [8].

The practice of MWHs is related with a multifaceted range of risk factors [9]. In SSA, the majority of births have been attended without a skilled healthcare provider [10]. Study revealed in Africa, MWHs may represent a useful strategy to improve prevention of mother to child transmission of HIV in high prevalence,and low-resource settings [11]. In 2019 Mini EDHS showed that 48% of live births were delivered in a health facility, and access to health facilities is mentioned to be more difficult in rural areas than in urban areas because of distance, scarce transport, and a lack of

appropriate facilities [12].MWH has several advantages [13,14]. It increases the use of skilled birth attendants [15], reduction maternal mortality [16] and avoid adverse pregnancy outcomes [15]. Despite these benefits, its utilization is low in sub-Saharan African countries [17–19] and there are numerous factors that influence the utilization of MWHs [20].In Ethiopia the introduction of MWHs service contributed to the 80% reduction in maternal mortality and still birth [21]. In addition, different literatures showed that facilities having MWHs for women with a risk of pregnancy-related complications had 47% and 49% lower risk of perinatal mortality and direct obstetric complication rate than facilities without MWHs, respectively [22]. The use of maternity waiting homes significantly contributed to an increase in the immediate uptake of postpartum family planning [23].WHO estimates that globally 81% of births were assisted by skilled health professionals between 2014–2019, ranging from 61% in sub-Saharan Africa to 99% in Europe, Central Asia and North America [24]. MWH helps to address first delay, the delay in deciding to seek care and the second delay, the delay to reach timely for obstetric care. So, MWH plays a great role in intervening those delays [25]. Maternal mortality remains a global issue particularly in developing countries and MWH is an important parts of the Sustainable Development Goals to reduce maternal mortality however its utilization is very low [26].

Globally, about 10.7 million women died in a year between 1990 and 2015 due to obstetric related cause [27]. Maternal death is 20 folds higher in developing countries than developed regions [28]. Maternal mortality is a global public health problem; maternal deaths were set at 211 maternal deaths per 100, 000 live births in 2017 [29]. MMR in SSA, is 415 per 100,000 live births, which is the highest in the world [30].Ethiopia is one of the Sub-Saharan African countries with a high MMR, 412 maternal deaths per 100,000 live births [31]. Developing counties accounted for approximately 99% of the estimated global maternal deaths, and Sub-Saharan Africa alone roughly accounted for 66% [32].Although there have been maternal waiting homes in Ethiopia for more than 30 years, they are inaccessible to the majority of pregnant mothers in rural areas [5]. Furthermore, studies show that women have a positive attitude to MWHs [33].Nevertheless, use of MWHs is still low utilization in most low income countries [9].In Ethiopia, almost 80% of its population reside in rural areas, where poor access to maternity services accounts for many maternal and perinatal deaths [34]. It has been evidenced that utilization of MWHS reduces maternal mortality by 80% and stillbirth rates by 73% in developing countries [35]. Ethiopia rests one of the nations with the top maternal death rates in the world. Although access to primary health coverage has increased from 50.7% in 2000 to more than 90% in 2019, the universal health coverage service coverage index remains at 43% [36].WHO recommended that the quality of evidence on utilization of MWHs is poor and insufficiently recognized. Further, additional research on "what strategies could be effective" in increasing utilization of MWHs and improving other key maternal and neonatal health outcomes [9]. Maternal delays in utilization of emergency obstetric care are the contributing factors for high maternal mortality in developing countries [37]. In middle and low income countries, low utilization of the MWH was due to distance from health facility structures of waiting home were identified as a principal barrier [20]. Rural women are around 4 times more likely to die because of pregnancy or delivery than women who came from urban areas with 95% CI [38].Levels of MWH utilization globally have been reported to be sub-optimal, relatively due to the poor quality of services available at MWHs [39].Most of the scholars focused on estimating the intention and knowledge of mothers to utilize MWHs for their current delivery even though ignorance of utilization [40]. In Ethiopia, the utilization of maternal waiting homes and its associated factors among women who gave birth in rural setting were not clearly described. However, no study was found during the literature review period that had been shown in the study area. Due to the above circumstances this study is designed to assess the utilization of maternal waiting homes among women who gave birth in rural setting to inspire planners and scholars to increasing maternal care services in Ethiopia.

## Methods and materials

### Study design, setting and period

Community based cross-sectional study design was employed. The study was conducted in Basona worena woreda from January to February, 2024 G.C. The governmental health center in the district has maternal waiting home, antenatal care,

delivery, and post-natal care services. Basona worena woreda is one of the city administrations in Debere Berhan, North shewa Zone, and Amhara regional state. It is located in the North of tarema bare, Southern Agolelanatera, East of Ankober and in the West of Mendida. The total area of the woreda is estimated to be 1185.63 sq. km.The total population size is about 100,521 as population projection calculation. Among them 49,255 are males and 51,266 are females and also it has 23,377 households. The woreda is currently includes 21 kebeles. The district has 3 governmental health center and 8 private clinics.

## Population

**Source population.** All households of rural area of Basona worena district that hosts women who gave birth in the past one year.

**Study population.** All selected households of rural area of Basona worena district kebeles or gotes that hosts women who gave birth in the past one year.

## Inclusion and Exclusion criteria

**Inclusion.** All Mothers who gave birth in the last one year and live in selected kebeles of the district during the data collection time were included.

**Exclusion criteria.** Mothers who were seriously ill during data collection period, and who lived in the selected kebeles for less than six months were excluded from the study.

## Sample size determination and Sampling procedures

A single population proportion formula was used for sample size calculation based on the assumptions for the proportion of MWHs utilization in Dabat District, North west, Ethiopia 16.2% [10] with 5% margin of error, 95% CI and considering 10% for non-respondent rate, design effect = 2, the sample size is increased to **460.**

In Basona worena district there are 21 rural kebeles and 198 Gotes; seven rural kebeles and 60 Gotes were selected from the sampling frame by simple random sampling. For each selected households, the sample was allocated proportionally to the numbers of mothers with respected to the households. When more than one eligible respondent was in the household, one respondent was randomly selected by a lottery method. Finally, 460 study participants were selected by multi-stage sampling technique seen as (Fig 1).

**Variables** described as (Fig 2).

## Dependent variable

◦ Utilization of maternal waiting home

## Independent variable

**Socio-demographic related factors.**

• Age of mother

• Marital status

• Educational status

• Occupation

• Monthly income

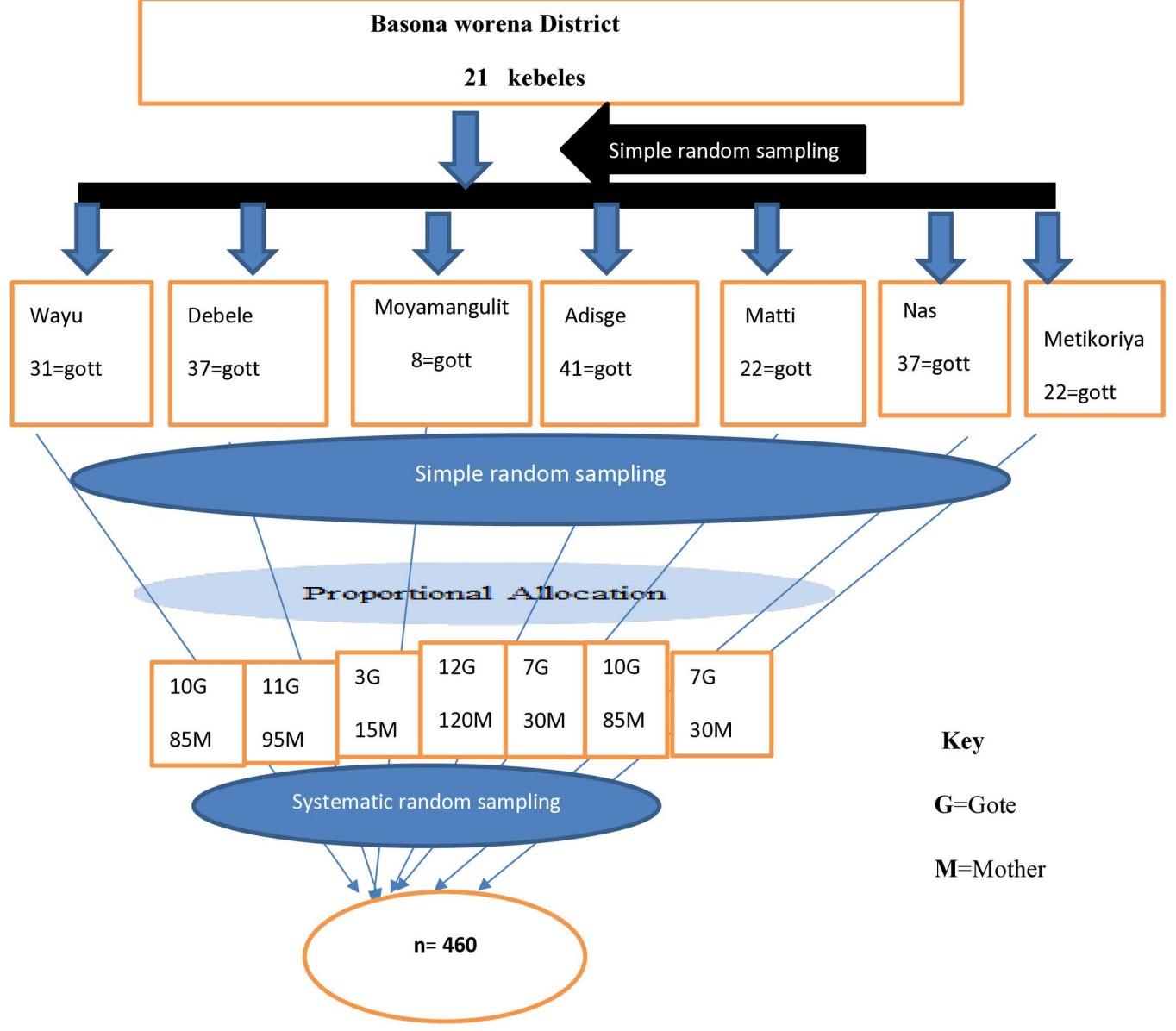

**Fig 1. Schematic presentation of sampling procedure for utilization of maternal waiting homes and associated factors among women who gave birth in rural setting of Basona worena woreda districts, Ethiopia, 2024.**

- Family size
- Having under five child's

**Obstetric related factors**

- Gravida
- ANC visit

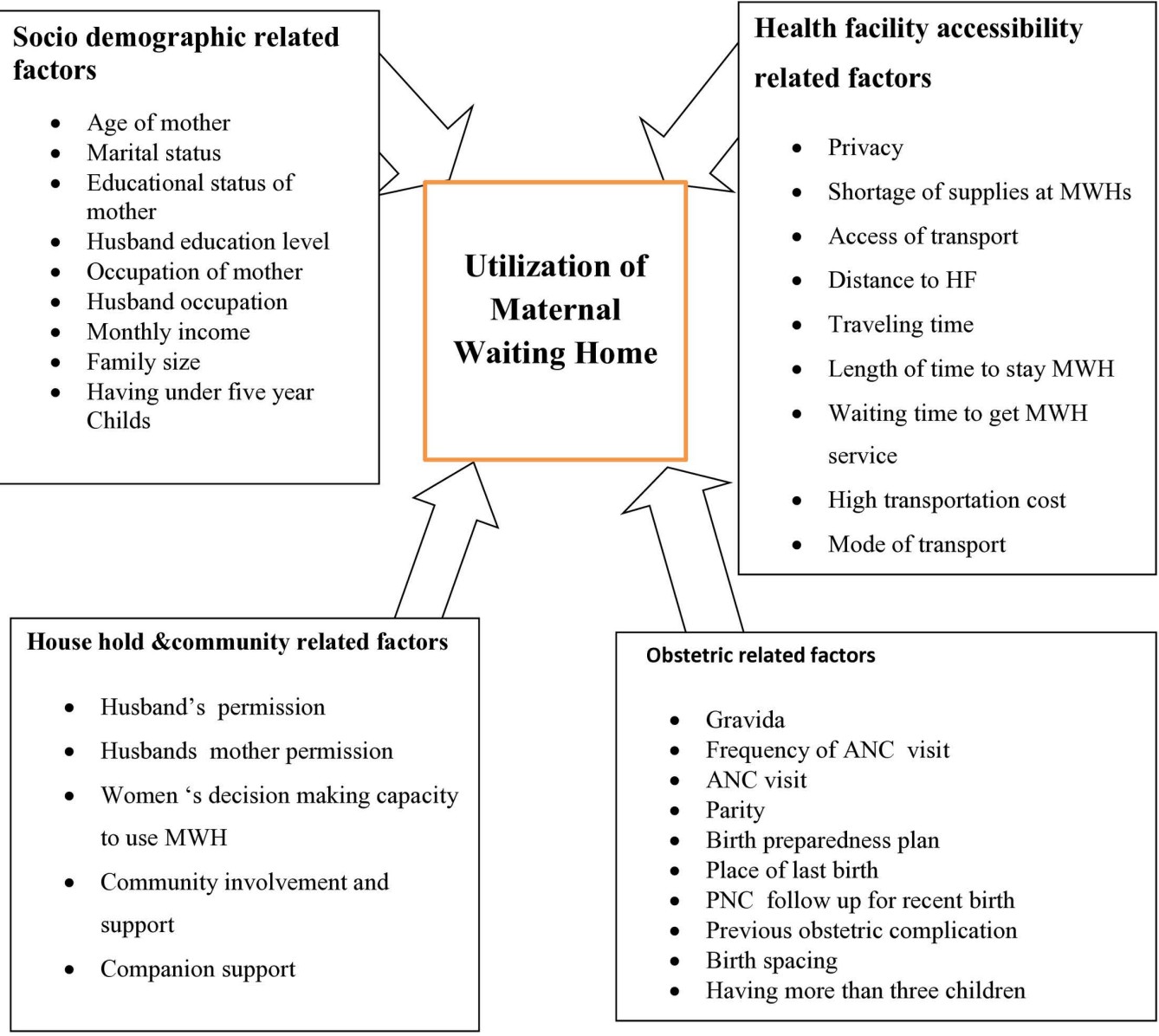

**Fig 2. The conceptual frame work of utilization of maternal home and its associated factors among women who gave birth in rural setting in Basona Worena District, 2024).**

• Frequency of ANC visit

• Parity

• Birth preparedness plan

• Place of last birth

• PNC follow up

- Previous obstetric complication

- Birth spacing

- Having more than three children

**Health facility accessibility related factors**

- Privacy

- Shortage of supply at MWH

- Modes of transport

- Distance to HF

- Travel time

- Waiting time to get MWH service

- Length of time to stay MWH

- Access of transport

- Transportation cost

**House hold and community related factors**

- Husband's permission

- Husband's mother permission

- Women's decision making capacity to use MWH

- Community involvement and support

- Companion support

**Operational and term definitions**

**Utilization:** those mothers who stay in MWH in the last (1–3) weeks of their pregnancy period.

**Maternal waiting home:** is a place for pregnant women to await birth in their last weeks (1–3 weeks) of pregnancy, close to emergency obstetric care [27].

**Previous obstetric complication:** refer to disruptions and disorders of pregnancy, labor and delivery, and the early neonatal period.

**Decision-making power:** Are husband and wife sitting down to discuss and decide about preparations for service utilization (yes or no) [41].

**Companion support:** Women were asked if they had someone to accompany them to health facility visits (yes or no) [42].

**Travel time to MWH:** The time it takes the pregnant woman to arrive at the nearby MWH when traveling on foot And it was considered "fair" if it is equal and less than 1h and "distant" if takes more than 1h on foot [43].

**Community involvement and support:** Women were asked if their community involve and support MWHs establishment (yes or no) [44].

## Data collection tool and technique

Structured and pretested interviewer-administered questionnaire was used to collect data. The questionnaires included four sections: Socio-demographic related factors, obstetrics related factors, health facility accessibility related factors and house hold and community related factors.

The questionnaire was first developed in English version and translated in to local Amharic language and reviewed by language experts for consistence of translation of the language before data collection. The study tool was prepared by adapting different related literatures with cronbachs alpha 0.98 [10]. Data were collected by four BSc midwifes and data collector was supervised by one MSc nurse and principal investigator.

## Data quality assurance

The questionnaire was reviewed by language experts for consistency of grammars and adapts literatures to check its appropriateness for assessing utilization of maternal waiting homes. The data were collected after 5% of the samples pretest was conducted. Then uncertain questions were corrected and redundant questions were excluded based on the pretest. The investigators and supervisors had day-to-day supervision throughout the whole period of data collection. Data collectors were trained for two days on the data gathering process to have a mutual understanding. Demonstration of interview was done for each data collectors to minimize error. The data collectors were closely supervised by the supervisors and principal investigator. Completeness of each questionnaire was checked by the principal investigator daily. Data consistency was tested by Cronbach's Alpha test (0.94).

## Data processing and analysis

Before analysis, data were first checked for completeness, clean, and coded. Data were entered to Epi-data version 4.6.2 and exported to SPSS version 25 software for cleaning and statistical analysis. The dependent variable was recoded to dichotomous out come as mothers with not used MWH were coded as "0" and those mothers with used MWH were coded as "1". Normality of continuous data distribution was examined. Categorical variables had been described using frequency, table, and figures. Independent predictors were coded based on previous related studies. Multicollinearity between independent variables were checked using Variable Inflation Factor (VIF), and no significant (mean VIF = 1.28) colinearity was detected. Model goodness of fit was checked by Hosmer-Lemeshow test, and the final model was fitted (p-value = 0.601). Bivariable logistic regression analyses were used and Crude Odd Ratio (COR) with 95% CI will be computed to assess the association between each predictor and the outcome variables. Variables with a p-value <0.25 during the bivariable analysis were included in the multi-variable logistic regression analysis. Multi-variable logistic regression analysis was conducted to identify the association between dependent and independent variables. Adjusting odds ratio (AOR) with 95% CI was estimated to identify the associated factors. Finally, statistical significance was declared at p value less than 0.05.

## Ethical approval

Ethical clearance and approval were obtained from the institutional review board (IRB) of Asrat Woldeyes Health Science Campus. After obtaining permission from Basona worena district health office, written and oral consent was obtained from the study participants, after informing them all the purpose, benefits, and voluntary nature of the participation in the study. All information obtained from the study participants would be kept private and confidential. Codes and aggregates reporting were used to eliminate names and other personal identifiers of respondents throughout the study process to ensure anonymity.

## 5. Result

### 5.1. Socio-demographic characteristics of respondents

A total of 460 mothers took part in the study, with a response rate of 100%. The mean (±SD) age of the women was 31(±7.36) years. The mean (±SD) average family monthly income was 2154.5(±1091.7) Ethiopian birr. Two hundred thirty

(50%) of the mothers were house wife and 290 (63%) of their husbands were farmer. Two hundred twelve (46.1%) of mothers attend primary school, and 445 (96.7%) were married as described as Table (1).

## 5.2. Obstetrics characteristics of participants

More than half (59.3%) of the study participants had ANC contact in recent pregnancies and 37(8%) had greater than five children. Two hundred sixty four (57.4%) of the study participants had planned pregnancies. However, 175 (38%) of them were a history of home delivery. Majority (73.7%) of the study participants had information about MWH. The most common (88.4%) reason not used MWH was not supported by family members or other community seen at Table (2).

**Table 1. Socio-demographic characteristics of the study participants in Basona worena district, North Showa, Ethiopia, 2024 (n = 460).**

| Variable | Category | Frequency | Percentage |
|---|---|---|---|
| Age of respondent | <=20 | 46 | 10.0 |
| | 21-25 | 57 | 12.4 |
| | 26-30 | 111 | 24.2 |
| | 31-35 | 88 | 19.2 |
| | >=36 | 158 | 34.2 |
| Estimated monthly income | < 2000ETB | 303 | 65.9 |
| | 2001-4000ETB | 95 | 20.7 |
| | >= 4001 ETB | 62 | 13.5 |
| Family size | < 5 | 175 | 38.0 |
| | >=5 | 285 | 62.0 |
| Numbers of under 5 children | Only one | 322 | 70.0 |
| | Only two | 138 | 30.0 |
| Marital status | Married | 445 | 96.7 |
| | Single | 3 | 0.7 |
| | Widowed | 6 | 1.3 |
| | Divorced | 6 | 1.3 |
| Maternal educational status | Cannot read &write | 154 | 33.5 |
| | Able to read &write | 78 | 17.0 |
| | Primary school | 212 | 46.1 |
| | Secondary school | 16 | 3.5 |
| Husband educational status | Cannot read &write | 254 | 55.2 |
| | Able to read &write | 78 | 17.0 |
| | Primary school | 63 | 13.7 |
| | Secondary school | 45 | 9.8 |
| | College &above | 20 | 4.3 |
| Maternal occupation | Farmer | 59 | 12.8 |
| | Government employ | 45 | 9.8 |
| | Merchant | 76 | 16.5 |
| | Daily laborer | 28 | 6.1 |
| | Student | 22 | 4.8 |
| | House wife | 230 | 50.0 |
| Husband occupation | Farmer | 290 | 63.0 |
| | Government employ | 50 | 10.9 |
| | Merchant | 95 | 20.7 |
| | Daily laborer | 25 | 5.4 |

**Table 2. Reproductive health Characteristics of the participants among women who gave birth in Basona worena district, North showa, Ethiopia, 2024(n = 460).**

| Variable | Category | Frequency | Percentage |
|---|---|---|---|
| Number of pregnancy | <=4 | 375 | 81.5 |
| | >4 | 85 | 18.5 |
| Number of life birth | <=2 | 280 | 60.9 |
| | 3-4 | 143 | 31.1 |
| | >=5 | 37 | 8.0 |
| ANC visit for recent birth | Yes | 273 | 59.3 |
| | No | 187 | 40.7 |
| Number of ANC visit | <2 | 180 | 39.1 |
| | 2-3 | 74 | 16.1 |
| | >=4 | 206 | 44.8 |
| Birth preparedness plan for recent birth | Yes | 264 | 57.4 |
| | No | 196 | 42.6 |
| Delivery assisted | Doctor | 52 | 11.3 |
| | Nurse/midwifes | 225 | 48.9 |
| | Trained traditional birth attendant | 10 | 2.2 |
| | Untrained traditional birth attendant | 173 | 37.6 |
| Place of last delivery | Health facility | 267 | 58.0 |
| | Home | 175 | 38.0 |
| | On the way to health facility | 18 | 4.0 |
| PNC follow up for recent birth | Yes | 333 | 72.4 |
| | No | 127 | 27.6 |
| Heard of MWH | Yes | 339 | 73.7 |
| | No | 121 | 26.3 |
| Source of information | Health professionals | 229 | 49.8 |
| | Others | 231 | 50.2 |
| Reason to use MWH (N = 261) | To get healthy child | 239 | 91.5 |
| | To get healthy mother | 186 | 71.3 |
| | To prevent mortality/disease | 121 | 43.4 |
| | To get better health care | 219 | 83.9 |
| | To prevent obstetric complication | 191 | 73.2 |
| Services during stay (N = 261) | Latrine | 131 | 50.2 |
| | Bedding | 260 | 99.6 |
| | Health professional check up | 249 | 95.4 |
| | Electricity | 61 | 23.4 |
| | Meals | 216 | 82.8 |
| | Coffee | 195 | 74.7 |
| | Clean water | 118 | 45.2 |
| | Bathing | 62 | 23.7 |

*(Continued)*

**Table 2.** (Continued)

| Variable | Category | Frequency | Percentage |
|---|---|---|---|
| Reason not to use MWH (N = 199) | Absence of MWH | 167 | 83.9 |
| | Lack of information | 103 | 51.7 |
| | Husband not permitted | 44 | 22.1 |
| | Not supported by others | 176 | 88.4 |
| | Not providing food in MWH | 1 | 0.5 |
| | Having under 5 children | 31 | 15.6 |
| | MWH not comfortable | 190 | 95.5 |
| | Not problem in previous pregnancy | 172 | 86.4 |
| | Negligence | 135 | 67.8 |
| | Distance from home | 33 | 16.6 |

## 5.3. Health facility-accessibility characteristics of participants

One hundred fourty nine (57%) of the study participants had difficult access to transport from home to health facility for MWH services and, 233(50.1%) of respondents were stayed in health care facilities prior to birth. One hundred fourty (53.7%) of the study participants had greater than one hour take time in nearest health facility and 234 (89.7%) of mothers were waiting less than thirty minute to get MWH services as described in Table (3).

## 5.4. Household and Community characteristics of participants

Two hundred fourty three (52.8%) of respondents had decision their own to their health but two hundred seventy five (59.8%) of the study participants had discussed with their husbands about MWH. Two hundred ninety six (64.3%) of husbands were supported their wives while 309 (67.2%) of the study participants have MWH utilization plan as described in Table (4).

## Utilization of MWH

In this study, the magnitude of MWH utilization was 56.7% (95% CI:52.4,61.3) seen as (Fig 3).

## Factors associated with MHW utilization among participants

Data were analyzed using binary logistic regression analysis. Statistical associations were checked by 95% CI and odds ratio. Those variables which had a p-value less than 0.25 in the binary logistical regression analysis were eligible for multi-variable logistic regressions. Finally, the adjusted odds ratio was checked and the significant variables p value<0.05 were considered as associated factors for maternal waiting home utilization.

Those mothers whose age category was aged between 26–30 years old were 78% less likely to utilized MWH than those women whose age category was 36 and above (AOR = 0.22,95% CI:0.08,0.65). Similarly, the odd of utilizing MWH is 3.2 times higher among mothers attending primary school than no formal education (AOR = 3.20,95%, CI:1.40,7.32). On the other hand, mothers who work government employee were 88% less likely to utilize MWH as compared to women whose work were farmer (AOR = 0.12,95%,CI:0.03,0.44). Likewise, mothers whose family members greater equal to five were nearly 3 times more likely to utilize MWH compare to women whose family members were less than five (AOR = 2.76,95%, CI:1.27,5.99).

The odds of utilizing MWH among mothers who had birth preparedness plan in recent birth were more than 10 times the odds of not having birth preparedness plan in recent pregnancy (AOR = 10.23, 95%,CI:9.8,29.3). Lastly, the odds of utilizing MWH among mothers who had maternity waiting home utilization plan in recent pregnancy were nearly 7 times the odds of not having maternity waiting home utilization plan(AOR = 6.82,95%,CI: 2.7,17.3) as seen Table (5).

**Table 3. Health facility accessibility characteristics of the participants on maternity waiting home utilization among women gave birth in Basona worena district, North showa, Ethiopia, 2024 (n = 460).**

| Variable | Category | Frequency | Percentage |
|---|---|---|---|
| Access to transport (n = 261) | Easy | 112 | 43.0 |
| | Difficult | 149 | 57.0 |
| Time to take in nearest health facility | Less 60 minutes | 121 | 46.3 |
| | Greater than 60 minute | 140 | 53.7 |
| Waiting time to get MWH services | Less than 30 minute | 234 | 89.7 |
| | Greater than 30 minute | 27 | 10.3 |
| Transport cost | Fair | 134 | 51.3 |
| | Not fair | 127 | 48.7 |
| Distance to health facility or MWH | Less than 5 km | 21 | 4.6 |
| | 5-10 km | 81 | 17.6 |
| | 10-15 km | 82 | 17.8 |
| | 15-20 km | 51 | 11.1 |
| | 20-25 km | 26 | 5.7 |
| Stayed in MWH | < 15 days | 233 | 50.1 |
| | >15 days | 28 | 6.1 |

**Table 4. House hold and community characteristics of the participants on maternity waiting home utilization among women who gave birth in Basona worena district, North showa, Ethiopa,2024 (n = 460).**

| Variable | Category | Frequency | Percentage |
|---|---|---|---|
| Discussion with husband | Yes | 275 | 59.8 |
| | No | 185 | 40.2 |
| MWH utilization plan | Yes | 309 | 67.2 |
| | No | 151 | 32.8 |
| Husband's mother support/permission | Yes | 269 | 58.5 |
| | No | 191 | 40.5 |
| Decision on maternal health | Husband | 200 | 43.5 |
| | Women (herself) | 243 | 52.8 |
| | Heath extension worker | 17 | 3.7 |
| Importance of MWH | Yes | 355 | 77.2 |
| | No | 105 | 22.8 |
| Husband permission/support | Yes | 296 | 64.3 |
| | No | 164 | 35.7 |
| Community involvement and support | Yes | 176 | 38.3 |
| | No | 284 | 61.7 |
| Companion support | Yes | 244 | 53.0 |
| | No | 216 | 47.0 |

## Discussion

In this study, the overall utilization of maternity waiting home was 56.7% (95% CI: 52.4, 61.3). This study is consistent with study conducted in Merit sub-city, isiolo country (61.1%) [45],in Somaliland (58%) [46], in Hadiya Zone, Southern Ethiopia (55.6%) [47].In the contrary, the finding of this study was lower than study conducted in rural Zambia (76.8%) [48]. The discrepancy might be explained by the variation in the sample size, socio-demographic characteristics and, as well as

# Magnitude of MWH utilization

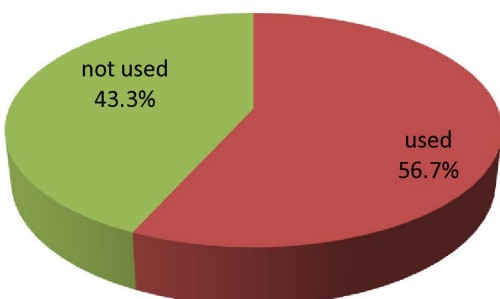

**Fig 3. Magnitude of MWH utilization among women who gave birth in Basona worena district, North showa, Ethiopia, 2024(n = 460).**

the difference in societal background and were used institutional based study method. The finding of this study also lower than study conducted in Sidama Zone, Southern, Ethiopia (67.5%) [49], East Welega Zone (65.3%) [50], SehalaSeyemit district, Waghimra Zone, (62.3%) [51]. The difference might be due to the variance in reliable promotion of maternity waiting home services for pregnant mothers until the expected date of delivery. On the other hand, due to inadequate birth preparedness and complication plans among pregnant women, and also, the preference to use a maternity waiting home can vary between and within geographical regions [40].

In this study, the magnitude of maternity waiting home utilization was higher than study conducted in Tanzania (31%) [52], GomoGoffa zone, Southern Ethiopia (48.8%) [28], Keffa Zone (42.5%) [53], Benchi-maji Zone(39%) [54], Jimma(38%) [41], Teltelle district(26.64%) [55], Arsi Zone, western Oromia (23.6%) [1], Finfinnee special zone, central Ethiopia (34%) [56], Southern region (16.7%) [2], and Dabat district, Northern Ethiopia (16.2%) [10]. The discrepancy might be due to the mobilization of health extension workers and the women's health development army in advocating, counseling, and advising maternal health services that are supported by the woreda health office. Further, explanation for the higher proportion might be the time variation. Nowadays, maternal health is a global priority area, and special focus might be given to increasing MWH utilization. This finding also higher than compare to studies conducted in rural Zambia (27.35%) [57], Kenya (10%) [19]. The variation might be due to social-background, cultural, economic status, and differences in small sample size and study period variation.

The study revealed that women's attending primary level of education is significantly associated with MWH utilization. This finding is consistent with the study conducted in Kenya ISIOLO district, rural Kenya, Butajira town, Ethiopia, Dabat district, North West, Ethiopia [10,19,45,58] respectively, which showed that educated mothers were more likely utilized MWH. The plausible reason might be that educational level increased awareness of health services, likely hoods of risk perception, level of understanding of new health-related information, easy acceptance of information and advice given by health care professionals, as well as better communication with their husbands, and having more decision-makers for their health that increased self-worth and confidence to care for their pregnancy. As a result, educated women will take care of their health and pregnancy. On the other hand, this is incongruent study conducted in rural Zambia, Tanzania, Jimma Zone, South west Ethiopia [3,42,57]respectively. This discrepancy may be due to difference socio-economic status, sample size, most of them were done facility based studies and variation in study period.

Another relevant finding in this study that those mothers whose age category was aged between 26–30 years old were significantly associated with MWH utilization. This finding is in line with study conducted in Tanzania, Southern Ethiopia,

**Table 5. Associated factors of maternity waiting home utilization among women who gave birth in Basonaworena district, North showa, Ethiopia, 2024 (n = 460).**

| Variable | Category | MWH Utilization | | COR (95% CI) | p-value | AOR 95% CI | P-value |
|---|---|---|---|---|---|---|---|
| | | Yes 56.7% | No 43.3% | | | | |
| Maternal age | <=20 | 37 | 9 | 4.11(1.98,8.51) | 0.000 | 0.34(0.06,1.92) | 0.219 |
| | 21-25 | 39 | 18 | 2.16(1.24,3.78) | 0.007 | 0.25(0.08,0.82) | 0.23 |
| | 26-30 | 60 | 51 | 1.176(1.01,1.70) | 0.04 | **0.22(0.08,0.65)** | **0.006**\*\* |
| | 31-35 | 49 | 39 | 1.26 (0.82,1.91) | 0.207 | 0.56(0.19,0.09) | 0.288 |
| | >=36 | 76 | 82 | 1 | | | |
| Estimated monthly income | <=2000 ETB | 178 | 125 | 1 | | | |
| | 2001-4000ETB | 60 | 35 | 1.71(1.13,2.60) | 0.011 | 0.69(0.28,1.67) | 0.41 |
| | >=4001ETB | 23 | 39 | 0.59(0.35,0.98) | 0.045 | 0.09(0.03,0.26) | 0.51 |
| Family size | < 5 | 114 | 61 | 1 | | | |
| | >=5 | 147 | 138 | 1.86(1.37,2.55) | 0.000 | **2.76(1.27,5.99)** | **0.01**\* |
| Maternal education | Can't read &write | 75 | 79 | 1 | | | |
| | Able to read &write | 50 | 28 | 1.76 (1.12,2.84) | 0.014 | 2.67(0.96,7.38) | 0.06 |
| | Primary school | 128 | 84 | 1.52(1.56, 2.01) | 0.003 | **3.20(1.40,7.32)** | **0.006**\*\* |
| | Secondary school | 8 | 8 | 1(0.38, 2.66) | 1.00 | 0.56(0.02,13.2) | 0.73 |
| Husband education | Can't read &write | 146 | 108 | 1 | | | |
| | Able to read &write | 48 | 30 | 1.6(1.04,2.53) | 0.04 | 0.07(0.02,0.22) | 0.10 |
| | Primary school | 28 | 35 | 0.8(0.48,1.32) | 0.37 | 0.06(0.01,0.20) | 0.11 |
| | Secondary school | 23 | 22 | 1.04(0.58,1.83) | 0.18 | 0.30(0.07,1.23) | 0.09 |
| | College &above | 16 | 4 | 4 (1.33,11.96) | 0.013 | 3.27(0.3,35.3) | 0.33 |
| Maternal occupation | Farmer | 26 | 33 | 1 | | | |
| | Gov't employed | 21 | 24 | 0.87 (0.48,1.57) | 0.000 | **0.12(0.03,0.44)** | **0.01**\* |
| | Merchant | 43 | 33 | 1.30(0.83, 2.05) | 0.203 | 0.29(0.08,1.03) | 0.06 |
| | Daily laborer | 12 | 16 | 0.75(0.35,1.58) | 0.045 | 0.18(0.04,0.92) | 0.40 |
| | Student | 10 | 12 | 0.83(0.36,1.92) | 0.07 | 0.08(0.02,0.53) | 0.80 |
| | House wife | 149 | 81 | 1.84(1.40,2.411) | 0.000 | 1.09(0.36,3.3) | 0.88 |
| Birth preparedness plan in recent birth | Yes | 210 | 54 | 3.89(2.88,5.24) | 0.000 | **10.23(6.9,29.3)** | **0.000**\*\*\* |
| | No | 51 | 145 | 1 | | | |
| MWH utilization plan in recent pregnancy | Yes | 236 | 73 | 3.23(2.48,4.20) | 0.000 | **6.82(2.7,17.3)**\* | **0.000**\*\*\* |
| | No | 25 | 126 | 1 | | | |
| Number of under five children | Only one | 233 | 89 | 1 | | | |
| | Only two | 28 | 110 | 0.25(0.16,0.38) | 0.000 | 0.42(0.02,0.09) | 0.10 |
| Number of pregnancy | <=4 | 215 | 160 | 1 | | | |
| | . > 4 | 46 | 39 | 1.17(0.7,1.80) | 0.231 | 0.65(0.24, 1.73) | 0.39 |
| Importance of MWH | Important | 253 | 102 | 2.48 (1.97,3.12) | 0.000 | 1.83(0.68,4.880) | 0.225 |
| | Not important | 8 | 97 | 1 | | | |

Significant at p-value <**0.05**\*, <**0.01**\*\*, <**0.001**\*\*\* COR = Crude odd ratio, **AOR** = Adjusted odd ratio.

Jimma Zone, West southern Ethiopia, Dabat district, North west, Ethiopia [10,41,59,60] respectively. This might be due to the fact that aged mothers might have matured children, which may have overtaken the general household activities. In addition, those older mothers might have had past obstetric practice and be concerned about a repetition of history by utilizing MWH. Also, older women have a greater chance of visiting health institutions and may get contacted by health care

professionals by getting sufficient information about maternity health services, including MWH. Moreover, older women may have higher decision making autonomy in the household on maternal and child health issue [61], So that they will decide utilized every maternity health services, including MWH.

Moreover, this find that mothers who work government employee were 88% less likely to utilize MWH as compared to women whose work were farmer. This finding is congruent with study conducted in Gamo Gofa Zone, Southern Ethiopia [28]. The possible reason might be that those women who are government employees might have exposure to information and better insight about maternity waiting home utilization services as compared to housewife women. On the other hand, this is inconsistent study conducted in Jimma zone, Sidama Zone, Finfinnee special zone, central Ethiopia [42,49,56] respectively. This discrepancy may be due to difference socio-cultural characteristics, sample size; most studies were done facility based studies.

The other finding revealed that birth preparedness plan in recent birth was significantly associated with MWH utilization. This finding is consistent with study conduct in rural areas of Arbaminch Zuria district, Gamo Gofa zone [62].The plausible reason might be due to the fact that in rural areas,lack of transportation option and poor accessibility to roads are major hindrance to access to life saving obstetric care in case of emergency. Because of this, pregnant mothers who had prepared to give birth in health institution preferred to stay maternity waiting home until they were due for child birth. It is also likely that those women who practice birth preparedness plan received enough counseling from health care providers which might include the use of MWH services.

This study revealed that mothers whose family members greater than five were nearly 3 times more likely utilize MWH compare to women whose family members were less than five. This finding is consistent with study conducted in Gedeo zone, southern Ethiopia, Butajira town, [58,63] respectively. The plausible reason might be due to mothers who had two or more children were more likely use Ante natal care, birth preparedness plan, and complication readiness than who had one child. This supported study conducted in Ethiopia [64]. Another study conducted in India found that women who had one or more live births were more likely to use the service than women who had no live births [65]. This may be due to the fact that women have more children have experienced more difficulties during pregnancy and childbirth in the past. They may also be motivated to seek out maternal health services, including MWH services, because they may have previously had prenatal consultations.

Lastly, the odds of utilizing MWH among mothers who had maternity waiting home utilization plan in recent pregnancy were nearly 7 times the odds of not having maternity waiting home utilization plan. This finding was congruent with study conducted in Arbaminch Zuria district, rural Zambia [62,66] respectively. This might be due to the MWH utilization plan, which is one strategy to increase MWH utilization for pregnant women living in the closest health care facility. This supported by study in Zambia [67]. Moreover, pregnant women appeal close observing and attention from the health care staff while they stay in MWH, allowing for rapid referral when complications happen. MWH has the reasonable to serve a great number of women and contribute to the improvement of maternal and newborn outcomes.

## Conclusion

The overall maternity waiting home utilization was 56.7%, which is relatively low. Significant predictors of maternity waiting home utilization included maternal age (26–30 years), family size, government-employed women, birth preparedness plan, maternity waiting home utilization plan, and primary level maternal education. Therefore, improving maternal waiting home utilization may involve broadening a strategy to raise women's educational status, health education communication, counseling, and advice regarding a maternity waiting home utilization plan and a birth preparedness plan.

### Recommendation

**For policy makers.**

- Based on the findings, policymakers are urged to focus more on educating healthcare professionals about maternal health care services that enhance women's MWH utilization plans for pregnancy complications and their birth

preparedness plans. Expand employment and education opportunities for mothers that increase knowledge and comprehension of their health status.

**For Basona woreda health office.** The Basona worena district health office should make sure that mothers remain in maternity waiting homes by implementing a birth preparedness plan and an MWH utilization plan that includes regular prenatal care follow-up, sufficient counseling, and professional advice. Support and inform health extension agents so they can mobilize the community about maternal health care, including maternity waiting home utilization services.

**For researcher's.** Future researchers could conduct longitudinal studies to determine the cause-and-effect relationship. To do qualitative study on MWH utilization and its associated factors.

## Limitation of the study

This research has the drawback of a cross-sectional study. A qualitative method was not used to aid this study. Moreover, the primary outcome was focused on women's self-reported MWH use, which may be prone to recall and social desirability bias.

## Acknowledgments

We would like to thank the study participants, data collectors, and supervisors who were involved in this study and spent their valuable time responding to my study.

## Author contributions

**Conceptualization:** Endale Menkir Degife.

**Data curation:** Endale Menkir Degife.

**Formal analysis:** Endale Menkir Degife.

**Funding acquisition:** Abebe Nigussie Ayele, Eyosiyas Yeshialem, Abdurrahman Mahammed Ahmed, Taye Anbessie Teklemariam.

**Investigation:** Abebe Nigussie Ayele, Endale Menkir Degife, Eyosiyas Yeshialem, Abdurrahman Mahammed Ahmed, Taye Anbessie Teklemariam.

**Methodology:** Abebe Nigussie Ayele, Endale Menkir Degife, Eyosiyas Yeshialem, Abdurrahman Mahammed Ahmed, Taye Anbessie Teklemariam.

**Project administration:** Abebe Nigussie Ayele, Endale Menkir Degife, Eyosiyas Yeshialem, Abdurrahman Mahammed Ahmed, Taye Anbessie Teklemariam.

**Resources:** Abebe Nigussie Ayele, Endale Menkir Degife, Eyosiyas Yeshialem, Abdurrahman Mahammed Ahmed, Taye Anbessie Teklemariam.

**Software:** Abebe Nigussie Ayele, Endale Menkir Degife, Eyosiyas Yeshialem, Abdurrahman Mahammed Ahmed, Taye Anbessie Teklemariam.

**Supervision:** Abebe Nigussie Ayele, Endale Menkir Degife, Eyosiyas Yeshialem, Abdurrahman Mahammed Ahmed, Taye Anbessie Teklemariam.

**Validation:** Abebe Nigussie Ayele, Endale Menkir Degife, Eyosiyas Yeshialem, Abdurrahman Mahammed Ahmed, Taye Anbessie Teklemariam.

**Visualization:** Abebe Nigussie Ayele, Endale Menkir Degife, Eyosiyas Yeshialem, Abdurrahman Mahammed Ahmed, Taye Anbessie Teklemariam.

**Writing – original draft:** Abebe Nigussie Ayele, Endale Menkir Degife, Eyosiyas Yeshialem, Abdurrahman Mahammed Ahmed, Taye Anbessie Teklemariam.

**Writing – review & editing:** Abebe Nigussie Ayele, Endale Menkir Degife, Eyosiyas Yeshialem, Abdurrahman Mahammed Ahmed, Taye Anbessie Teklemariam.

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
