## [Decision Letter · Decision Letter 0]

25 Feb 2025

PONE-D-24-28442Maternity Waiting Homes Utilization and associated factors among women who gave birth in the last one year in rural settings of Basona Worena District, Ethiopia: A cross sectional Study.PLOS ONE

Dear Dr. Ayele,

Thank you for submitting your manuscript to PLOS ONE. After careful consideration, we feel that it has merit but does not fully meet PLOS ONE’s publication criteria as it currently stands. Therefore, we invite you to submit a revised version of the manuscript that addresses the points raised during the review process.

We look forward to receiving your revised manuscript.

Kind regards,

Ermel Johnson, MD, MPH, PhDc

Academic Editor

PLOS ONE

Journal Requirements:

 -https://doi.org/10.1038/s41598-023-39029-1

-https://doi.org/10.1371/journal.pone.0265182

-https://www.ajol.info/index.php/ejhd/article/view/232044

In your revision ensure you cite all your sources (including your own works), and quote or rephrase any duplicated text outside the methods section. Further consideration is dependent on these concerns being addressed.

“no competing of interest exist .”

5. Please provide a complete Data Availability Statement in the submission form, ensuring you include all necessary access information or a reason for why you are unable to make your data freely accessible. If your research concerns only data provided within your submission, please write "All data are in the manuscript and/or supporting information files" as your Data Availability Statement.

Reviewers' comments:

Reviewer's Responses to Questions

**Comments to the Author**

1. Is the manuscript technically sound, and do the data support the conclusions?

Reviewer #1: Partly

Reviewer #2: Yes

2. Has the statistical analysis been performed appropriately and rigorously? 

Reviewer #1: I Don't Know

Reviewer #2: Yes

3. Have the authors made all data underlying the findings in their manuscript fully available?

Reviewer #1: Yes

Reviewer #2: Yes

4. Is the manuscript presented in an intelligible fashion and written in standard English?

Reviewer #1: No

Reviewer #2: Yes

5. Review Comments to the Author

Reviewer #1: Dear editor, thank you so much for your invitation.

And also I appreciate the authors for their efforts they made this study. However, I do not recommend the publication of this study in the current status so the authors should correct and respond to comments and question mentioned hereunder

Abstract

“owever, little is known in Ethiopia on the utilization of maternity waiting home among actual mothers who gave birth in rural settings and their involvement of the service utilization is not well explored.” revise this sentence

“In Ethiopia, the utilization of maternal waiting homes and its associated factors among women who gave birth in rural setting were not clearly described.” redundancy of the very earliest sentence.

Please, correct the tense error from the objective

“(>=5)”would be written in clear term.

“The overall maternity waiting home utilization was 56.7% which relatively low compare to previous studies” grammatical error.

Introduction

This section must be revised. It lacks consistency and coherence.

“Most of the scholars focused on estimating the intention and knowledge of mothers to utilize MWHs for their current delivery” cite this.

“However, little is known in Ethiopia on the utilization of MWHs among actual mothers who gave birth in rural settings and their involvement of the service utilization is not well explored. In Ethiopia, the utilization of maternal waiting homes and its associated factors among women who gave birth in rural setting were not clearly described. However, no study was found during the literature review period that had been shown in Basona worena district.” revise these sentences too. Pay attention.

Methods and materials

Please, use universally acceptable expressions, where all scientists can understand. For instance, what is 2015 E.C.?

Variables (Figure 2)?

The authors would cite the source of their tool and append their Cronbach's Alpha test value

Results:

The percentages of observations of some variables are not correct.

Replace “Reproductive health Characteristics of participants” by Obstetrics characteristics of participants. Because reproductive health is a broad term or not limited to the characteristics you mentioned in this study.

The independent variables depicted in the variable section and the result section are inconsistent.

Replace “Multivariate logistic regression” by factors associated with MHW utilization among participants.

Line 288-292 is the redundancy of data analysis method. The result section should only include results.

“Those mothers whose age category was aged between 26-30 years old were 0.22 times more likely to utilized MWH than those women whose age category was 36 and above (AOR=0.22,95% CI:0.08,0.65).” Check this interpretation

Put the row percentages of yes and no observation of MWH utilization in table 5

Discussion

Note that family size and parity are different variables.

Studies limitations would be mentioned under this section and remove the strength you have mentioned. A sampling technique that was based on your study’s nature and design should not mention as study’s strength.

Conclusion

The conclusion seems like results. Not appropriately written

some the recommendations are not based on your findings

Ethical approval

How could you take a written informed consent form those 33.5% your study participants who were unable to read and write? This should be clarified.

General suggestions

Figures are not numbered correctly.

Omit meaningless arrows from the figure that illustrated sampling procedure

The manuscript has serious grammar and punctuation errors. So the authors must use language edition services.

Reviewer #2: I feel the authors have done all the necessary corrections suggested. This is a very important topic, as worldwide, rural areas do not have access to healthcare easily, and this leads to worse maternal and neonatal complications. By having maternal waiting homes, outcomes are expected to be improved.

6. PLOS authors have the option to publish the peer review history of their article (what does this mean? ). If published, this will include your full peer review and any attached files.

**Do you want your identity to be public for this peer review?** For information about this choice, including consent withdrawal, please see our Privacy Policy .

Reviewer #1: No

Reviewer #2: No

---

## [Author Response · Author response to Decision Letter 1]

1 Apr 2025

We are thoroughly corrected the comments of reviewers and we appreciate the reviewers and editors. thank you so much !!!!!

---

## [Editor Report · Decision Letter 1]

30 Apr 2025

PONE-D-24-28442R1Maternity Waiting Homes Utilization and associated factors among women who gave birth in the last one year in rural settings of Basona Worena District, Ethiopia: A cross sectional Study.PLOS ONE

Dear Dr. Ayele,

Thank you for submitting your manuscript to PLOS ONE. After careful consideration, we feel that it has merit but does not fully meet PLOS ONE’s publication criteria as it currently stands. Therefore, we invite you to submit a revised version of the manuscript that addresses the points raised during the review process.

We look forward to receiving your revised manuscript.

Kind regards,

Ermel Johnson, MD, MPH, PhDc

Academic Editor

PLOS ONE

Journal Requirements:

Reviewers' comments:

Abstract

“owever, little is known in Ethiopia on the utilization of maternity waiting home among actual mothers who gave birth in rural settings and their involvement of the service utilization is not well explored.” revise this sentence

“In Ethiopia, the utilization of maternal waiting homes and its associated factors among women who gave birth in rural setting were not clearly described.” redundancy of the very earliest sentence.

Please, correct the tense error from the objective

“(>=5)”would be written in clear term.

“The overall maternity waiting home utilization was 56.7% which relatively low compare to previous studies” grammatical error.

Introduction

This section must be revised. It lacks consistency and coherence.

“Most of the scholars focused on estimating the intention and knowledge of mothers to utilize MWHs for their current delivery” cite this.

“However, little is known in Ethiopia on the utilization of MWHs among actual mothers who gave birth in rural settings and their involvement of the service utilization is not well explored. In Ethiopia, the utilization of maternal waiting homes and its associated factors among women who gave birth in rural setting were not clearly described. However, no study was found during the literature review period that had been shown in Basona worena district.” revise these sentences too. Pay attention.

Methods and materials

Please, use universally acceptable expressions, where all scientists can understand. For instance, what is 2015 E.C.?

Variables (Figure 2)?

The authors would cite the source of their tool and append their Cronbach's Alpha test value

Results:

The percentages of observations of some variables are not correct.

Replace “Reproductive health Characteristics of participants” by Obstetrics characteristics of participants. Because reproductive health is a broad term or not limited to the characteristics you mentioned in this study.

The independent variables depicted in the variable section and the result section are inconsistent.

Replace “Multivariate logistic regression” by factors associated with MHW utilization among participants.

Line 288-292 is the redundancy of data analysis method. The result section should only include results.

“Those mothers whose age category was aged between 26-30 years old were 0.22 times more likely to utilized MWH than those women whose age category was 36 and above (AOR=0.22,95% CI:0.08,0.65).” Check this interpretation

Put the row percentages of yes and no observation of MWH utilization in table 5

Discussion

Note that family size and parity are different variables.

Studies limitations would be mentioned under this section and remove the strength you have mentioned. A sampling technique that was based on your study’s nature and design should not mention as study’s strength.

Conclusion

The conclusion seems like results. Not appropriately written

some the recommendations are not based on your findings

Ethical approval

How could you take a written informed consent form those 33.5% your study participants who were unable to read and write? This should be clarified.

General suggestions

Figures are not numbered correctly.

Omit meaningless arrows from the figure that illustrated sampling procedure

The manuscript has serious grammar and punctuation errors. So the authors must use language edition services.

---

## [Author Response · Author response to Decision Letter 2]

9 May 2025

thank you for valuable and constructive feed back. We are eager to answer your question and accept your valuable comments. We are strongly amend the comments and answer the question. Thank you!!!!!!!!!!!

---

## [Editor Report · Decision Letter 2]

10 Sep 2025

Maternity Waiting Homes Utilization and associated factors among women who gave birth in the last one year in rural settings of Basona Worena District, Ethiopia: A cross sectional Study.

PONE-D-24-28442R2

Dear Dr. Ayele,

We’re pleased to inform you that your manuscript has been judged scientifically suitable for publication and will be formally accepted for publication once it meets all outstanding technical requirements.

Kind regards,

Jianhong Zhou

Staff Editor

PLOS ONE
---

## [Editor Report · Acceptance letter]

PONE-D-24-28442R2

PLOS ONE

Dear Dr. Ayele,

I'm pleased to inform you that your manuscript has been deemed suitable for publication in PLOS ONE. Congratulations! Your manuscript is now being handed over to our production team.

Kind regards,

on behalf of

Dr. Jianhong Zhou

Staff Editor

PLOS ONE